# Preliminary Clinical Experience with a Novel Optical–Ultrasound Imaging Device on Various Skin Lesions

**DOI:** 10.3390/diagnostics12010204

**Published:** 2022-01-15

**Authors:** Gergely Csány, László Hunor Gergely, Norbert Kiss, Klára Szalai, Kende Lőrincz, Lilla Strobel, Domonkos Csabai, István Hegedüs, Péter Marosán-Vilimszky, Krisztián Füzesi, Miklós Sárdy, Miklós Gyöngy

**Affiliations:** 1Dermus Ltd., Sopron út 64, 1116 Budapest, Hungary; kiss.norbert@med.semmelweis-univ.hu (N.K.); lilla.strobel@dermusvision.com (L.S.); domonkos.csabai@dermusvision.com (D.C.); istvan.hegedus@dermusvision.com (I.H.); peter.marosan@dermusvision.com (P.M.-V.); krisztian.fuzesi@dermusvision.com (K.F.); miklos.gyongy@dermusvision.com (M.G.); 2Faculty of Information Technology and Bionics, Pázmány Péter Catholic University, Práter utca 50/A, 1083 Budapest, Hungary; 3Department of Dermatology, Venereology and Dermatooncology, Semmelweis University, Mária utca 41, 1085 Budapest, Hungary; gergely.laszlo.hunor@med.semmelweis-univ.hu (L.H.G.); szalai.klara@med.semmelweis-univ.hu (K.S.); lorincz.kende@med.semmelweis-univ.hu (K.L.); sardy.miklos@med.semmelweis-univ.hu (M.S.)

**Keywords:** dermatology, skin ultrasound, skin cancer, chronic skin inflammation, portable ultrasound, melanoma, basal cell carcinoma, seborrheic keratosis, dermatofibroma, naevus

## Abstract

A compact handheld skin ultrasound imaging device has been developed that uses co-registered optical and ultrasound imaging to provide diagnostic information about the full skin depth. The aim of the current work is to present the preliminary clinical results of this device. Using additional photographic, dermoscopic and ultrasonic images as reference, the images from the device were assessed in terms of the detectability of the main skin layer boundaries and characteristic image features. Combined optical-ultrasonic recordings of various types of skin lesions (melanoma, basal cell carcinoma, seborrheic keratosis, dermatofibroma, naevus, dermatitis and psoriasis) were taken with the device (N = 53) and compared with images captured with a reference portable skin ultrasound imager. The investigator and two additional independent experts performed the evaluation. The detectability of skin structures was over 90% for the epidermis, the dermis and the lesions. The morphological and echogenicity information observed for the different skin lesions were found consistent with those of the reference ultrasound device and relevant ultrasound images in the literature. The presented device was able to obtain simultaneous in-vivo optical and ultrasound images of various skin lesions. This has the potential for further investigations, including the preoperative planning of skin cancer treatment.

## 1. Introduction

As a result of continuous technical developments, various non-invasive imaging procedures are becoming increasingly adopted for the field of dermatology [1]. These techniques include ultrasound (US) imaging, which is playing an increasingly important role in dermatology [2].

Skin cancer is one of the most common types of cancers in developed countries. In the United States, it is the most common type of cancer, affecting 20% of the population in their lifetime [3]. Early and accurate screening, and appropriately chosen, planned and implemented treatments play a key role in preventing skin cancer deaths [4]. Inflammatory skin diseases such as atopic dermatitis and psoriasis are also common skin maladies. Atopic dermatitis affects roughly 10% of the population in developed countries [5,6]. A total of 40% of the patients affected have moderate to severe symptoms [7]. Ultrasound imaging has the potential to play an important role in improving the detection, treatment planning and follow-up of the above-mentioned skin diseases [8,9,10,11,12,13]. Combining the ultrasound information with optical imaging can further strengthen diagnostic accuracy [14], since ultrasound information alone may not be sufficient for the diagnosis of certain skin diseases such as skin cancer [15].

Ultrasound imaging makes it possible to map and objectively examine the subsurface structures of human tissues in a safe and cost-effective manner. Compared to more established ultrasound applications such as abdominal imaging, skin ultrasound uses higher frequencies (>15 MHz), with the depth of penetration posing a limit to the frequency that can be used. At a frequency of around 20 MHz, a depth of 6–7 mm and a resolution of 50–200 μm can be achieved [16]. High-frequency ultrasound imaging has the potential to improve the accuracy of the diagnosis of certain skin diseases [17,18,19] as well as the planning of different interventions and the monitoring of treatment results [20,21]. Due to the above advantages and applications, the use of ultrasound in dermatology is emerging [22]. However, due to the cost of the above-mentioned high-frequency devices currently available and the difficulty of interpreting skin ultrasound images, it is mostly only available in larger medical centers. Given the workload on the centers, which may have month-long waiting times [23], it would be useful if dermal ultrasound technology were more widely available. It is essential for the above aim to reduce the cost, increase portability, and improve and extend the interpretability of the images for non-radiologist users of skin ultrasound imaging devices [24].

A recent paper by Mlosek et al. [25] provides a valuable review of current skin ultrasound devices, noting that dedicated skin ultrasound scanners generally have higher portability than classical ultrasound scanners. Among the dedicated skin ultrasound scanners, examples of commonly used systems are the Atys Dermcup [26], Cortex DermaScan C [10], Clarius L20 HD [27], Dramiński DermaMed [28], Episcan I-200 [29,30], Vevo MD [31]. In general, the devices have varying degrees of portability, and none of them provide a viewfinder that shows in real time where the ultrasound image is taken on the skin. The aim of developing the *Dermus SkinScanner* was to address this need while also ensuring a high degree of portability and usability. The *Dermus SkinScanner* is currently a premarket device, with a novel feature of combining optical and ultrasound imaging in an integrated, handheld device. This feature aids the precise positioning of the ultrasound recordings, which in turn potentially aids reproducibility of the examinations, an aim expressly stated in a recent position statement of the European Federation of Societies for Ultrasound in Medicine and Biology (EFSUMB) on dermatologic ultrasound (Position Statement 5) [22].

## 2. Materials and Methods

### 2.1. The Dermus SkinScanner Device

The *Dermus SkinScanner* is a wireless, compact portable optical and ultrasound imaging device that is developed by Dermus Ltd. (Budapest, Hungary). The compact device contains the imaging, data processing and display systems, together with its own battery-based power supply (Figure 1). It is easy to handle, relatively light-weight (470 g) and provides a user interface including the display of recordings at the point of the examination (Figure 1). The *Dermus SkinScanner* also contains an optical imaging module that helps the positioning, and thus also the reproducibility, of ultrasound imaging.

The *Dermus SkinScanner* uses a single-element ultrasound transducer with 33 MHz nominal center frequency. Two-dimensional ultrasound imaging is realized via mechanical scanning applying a physical slider on the device and an automated scan conversion algorithm [32], verified using phantoms and dermatological data in previous work [33]. The imaging window is covered by a silicone membrane. Ultrasound gel is applied on the membrane surface for scanning.

As mentioned above, a unique feature in comparison to other portable ultrasound imaging devices is that the *Dermus SkinScanner* uses optical guidance for the enhancement of precise positioning and repeatability of the recordings. In practice, this means that real-time optical images of the area of scanning are displayed with 10-fold magnification during positioning, before scanning. The results of a single recording are then displayed in terms of a 2-D ultrasound image and the corresponding optical image of the surface, with a marker indicating the relative position of the ultrasound image slice to the surface. In order to provide higher contrast and facilitate detection of skin structures, the ultrasound image employs a color scale instead of a grayscale representation. The field of view of the optical image is 15 mm × 15 mm. The ultrasound image extends 12 mm laterally, with a maximum penetration depth of 10 mm. The image acquisition time is two seconds. Both the optical and ultrasound images of the device are automatically and anonymously saved using a cloud system.

### 2.2. Study Design

The present study was performed under the ethical approval of the Hungarian National Institute of Pharmacy and Nutrition (approval number OGYÉI/8317/2020), with all subjects providing their written informed consent. The examination site was the Department of Dermatology, Venereology and Dermatooncology, Semmelweis University, Budapest, Hungary. The examinations were performed by a dermatologist professional, hereinafter referred to as the investigator.

#### 2.2.1. Study Population

The study included patients with a relatively wide range of skin lesions. The patients were primarily chosen from daily outpatient care, based on the criteria of the presence of skin lesions that were affecting the dermis and/or epidermis and that could be examined with ultrasound, and of being exempt from the exclusion criteria. Exclusions were limited to the presence of any signs of damaged, bleeding or purulent skin surfaces or skin lesions in the area to be examined, the location of the lesion being over an eyelid.

The patient population included both women (38%) and men (62%), with mean ages of 53.7 ± 21.9 and 54.1 ± 19.5 years, respectively. In total, 53 lesions of 39 patients were examined. The vast majority (91%) of the lesions were benign lesions or malignant skin tumors (including two hemangiomas, one dermatofibroma, 17 naevi, five keratoses, 12 basal cell carcinomas, two squamous cell carcinomas, eight melanomas and one cutaneous metastasis of melanoma) and a smaller proportion (9%) consisted of inflammatory skin lesions (one dermatitis, two psoriasis, one morphea and one abscess). Regarding the locations of the lesions, the skin of several body regions has been examined, including those of the head, face, nose, upper and lower limbs, chest, abdomen and back.

#### 2.2.2. Data Acquisition

Several optical and ultrasound images were taken of each lesion during the examinations. First, a clinical (macroscopic) and a dermoscopic photo (*DermLite DL100*, polarized, non-contact mode, 3Gen Inc., San Juan Capistrano, CA, USA) were taken of the skin lesion. This was followed by ultrasound recordings using two different portable skin imaging devices: the *Dermus SkinScanner* (as described in Section 2.1) and a reference device, the *Dramiński DermaMed* (briefly described below).

The *DermaMed* (DRAMIŃSKI S.A., Olsztyn, Poland) is a light-weight (300 g), high-frequency (dual-mode 30/48 MHz, used in the 48 MHz mode) ultrasound imaging device that requires a USB cable connection to a PC that is used for power supply and for the processing, display and storage of two-dimensional grayscale ultrasound images (or a series of such images). The probe uses tilting mechanical scanning with a single-element transducer moved by an electrical motorized scanning system in a chamber that should be filled with distilled water and closed by a stretched film by the user, for appropriate coupling.

As described in Section 2.1, the optical image of the *Dermus SkinScanner* stored the information of the spatial position and orientation of its ultrasound scans in relation to the surface of the skin area of examination. This information was not available for the *Dramiński DermaMed* recordings. Nevertheless, they were taken with a technique that kept the spatial orientation of its scans as close to those of the other device as possible. However, a perfectly precise alignment was not possible due to the probe physically covering the area of examination during scanning.

To present the results of examinations of specific cases, the corresponding figures show the images (of various imaging modalities and devices, as described above) in a consistent layout. The layout is presented below in Figure 2. The primary layers associated with the skin (epidermis, dermis, subcutis) and the lesions themselves are indicated by certain markers as an aid for the interpretation of the images.

#### 2.2.3. Evaluation of Skin Structure Detectability

The ultrasound images taken with the *Dermus SkinScanner* were evaluated and compared to those obtained with the *Dramiński DermaMed*, as a reference, in terms of the detectability of structural elements of the skin, such as the epidermis, the dermis, the subcutis and the lesions. The evaluation was performed by both the investigator (on site) and by two independent experts (off site).

Since the use of the *Dermus SkinScanner* device aims to exploit the novel field at the intersection of dermatology and radiology, the two independent experts were chosen for the evaluation as representatives from these two medical specialties to cover the interdisciplinary field of dermatologic ultrasonography. The independent experts included a dermatologist with limited experience and a radiologist with extensive experience (10+ years) in skin ultrasound imaging. The two independent experts did not participate in any of the examinations and also did not have previous experience with the ultrasound devices used in the study.

## 3. Results

Following a summary of skin structure detectability and a brief section on interpreting the *Dermus SkinScanner* optical–ultrasound images (Figure 3), representative images of various common skin lesions were presented. For the cases shown in Figure 4a,b, Figure 5b,c and Figure 6c, a histopathological diagnoses were also available, which in all cases confirmed the preliminary diagnosis of the investigator.

### 3.1. Skin Structure Detectability

The results for the detectability of different structures (lesions, epidermis, dermis, subcutis) of the skin, as evaluated by both the investigator and by two independent experts (Section 2.2.3), are summarized in Table 1. Please note that the only necessary standard of reference was that all imaged regions contained all four skin structures of interest.

On average, the detection performance of the *Dermus SkinScanner* device was 93.0%, while that of the reference device was 94.7%, showing the *Dermus SkinScanner* to be non-inferior within a margin of 5% with a confidence interval of 95% [34]. The detectability of the lesions seemed slightly better for the *Dermus SkinScanner* images, while lesion detectability on the skin layers seemed superior for the reference device. This was in line with the overall experience description given by the evaluators, where they concluded that the *Dermus SkinScanner* images generally provided better contrast of the lesions with clearer borders than the reference device, which was generally more applicable for imaging relatively larger lesions (with >1.5 cm diameter).

The investigator reported being able to distinguish both the lesions and the skin layers on the vast majority (92%) of the ultrasound images of both devices. In only 4% of the cases, he reported being unable to distinguish any of these structures on both images (in the cases of a large lesion below the nose, and of a thin lesion directly next to a toenail). In another 4% of the cases, he was able to distinguish certain structures with higher confidence on the *Dermus SkinScanner* images.

Comparison of the evaluations showed a relatively high correspondence (with differences within 10%) for the two independent evaluations in addition to the on-site evaluations. The results suggest that the interpretation of the images of the *Dermus SkinScanner* was slightly aided by relatively minimal training and experience for a non-radiologist user with knowledge of the skin (e.g., a dermatologist).

### 3.2. Dermus SkinScanner Ultrasound Image Interpretation

The *Dermus SkinScanner* displays 2-D ultrasound images with a color code for a greater contrast between the structural elements in the images. A dark color represents the lowest intensity which is followed by green and blue up to red and yellow which stands for the highest intensity value.

Regarding the structures of the skin, ultrasound reflections from the epidermis usually appear as a high-intensity, thin structure in the images. The dermis appears as a thicker structure with inhomogeneities in ultrasound reflection intensity. The subcutis usually appears as a low-intensity region with scattered structures in it appearing with a higher intensity. This is consistent with the literature on other dermatological ultrasound images [2].

Typical elements of the ultrasound images of the *Dermus SkinScanner* are presented in Figure 3 using representative images of a cutaneous inflammation and of a cutaneous neoplasm. The spatial location of an ultrasound image slice is indicated by the red line on the corresponding optical image of the surface.

It should be noted that several artefacts may appear in the images besides the structures of interest. These artefacts include the ultrasound reflections from the outer structures and from the membrane of the imaging window, possible air bubbles from the the gel and consequent acoustic shadows in the image of the deeper structures below the bubbles. Regarding the reflections related to the imaging window, the artefacts are planned to be removed or at least substantially reduced for an easier interpretation of the images in the next generation of the device. Regarding air bubbles, the related artefacts can be reduced or eliminated with special attention when applying the gel on the surface of the imaging window.

### 3.3. Melanoma

Figure 4a presents a typical case of melanoma. Clinically, on the surface, the lesion appears with asymmetric shape and diverse colors, with a central elevated part. The dermoscopy shows a disorganized pattern, asymmetric pigmentation with multiple colors, a blue-whitish veil and atypical vessels. Regarding the ultrasound images, the characteristic hypoechoic spindle-like depth morphology of melanoma is clearly observable in Figure 4a. The thickness of the lesion was measured to be 1.97 mm on the ultrasound image of the *Dermus SkinScanner*. This is slightly higher than the 1.68 mm thickness measured by histopathology, as expected due to tissue shrinkage after excision and during histological preparation [35]. The thickness measured on the reference ultrasound image, however, is smaller than both the histopathological and *Dermus SkinScanner* measurements: 1.17 mm. This could be due to the difficulty of imaging the right cross-section of the lesion on this image without an optical aid.

It is significantly more difficult to identify a cutaneous metastasis of melanoma since it has no characteristic clinical and dermoscopic features. However, Figure 4b shows that its presence is clearly noticeable and localizable on the ultrasound images. It appears as a hypoechoic, heterogeneous lesion in the deeper parts of the dermis, with irregular borders and asymmetric shape. These observations are in correspondence with those found in the literature [36].

### 3.4. Basal Cell Carcinoma

Basal cell carcinomas (BCC) may not show clear borders as seen from the surface of the skin, therefore skin ultrasound examination can help in clarifying these borders [21,37].

In Figure 5a, a BCC is present optically as a shiny, slightly elevated yellow-whitish papule, and the dermoscopy shows arborising vessels and shiny white blotches. Regarding the ultrasound images, a relatively compact but asymmetric morphology was present (being distinguishable from the spindle-like morphology of Figure 4a). A lesion is generally more hypoechoic than the surrounding dermis, but echoes of ultrasound signals can be observed within the lesion (it is less hypoechoic than a typical melanoma). It is observable on both ultrasound images that the skin area presented in this case contained a dermis that was relatively highly hyperechoic.

Figure 5b presents a clinically flat, shiny erythematosus lesion, with the dermoscopy showing arborising vessels, shiny white blotches and signs of ulceration. The ultrasound images show a relatively hypoechoic lesion with a more hypoechoic central region and fewer hypoechoic boundary regions, and with an irregular morphology.

There are body locations in which it is more difficult to perform ultrasound imaging, such as inclinations and curves of the nose, ears, eye area, mouth area. It is presented below in Figure 5c that it is possible to record relatively good quality images even in those areas.

The BCC presented in Figure 5c is clinically a shiny, slightly sunken, scaly lesion. Dermoscopy shows arborising vessels and ulcerations. A heterogeneously hypoechoic lesion with a more hypoechoic center and less hypoechoic boundary regions with border irregularities is present in the dermis on the ultrasound images. The lesion borders were evidently more detectable on the *Dermus SkinScanner* images as compared to the reference image in this case.

### 3.5. Seborrheic Keratosis

Figure 6a depicts a clinically slightly elevated, brownish keratotic papule. In comparison, Figure 6b depicts a clinically asymmetric shape and color, with a slightly elevated, keratotic surface, with the dermoscopy showing sharply demarcated borders, milia-like cysts, fissures and ridges, and network-like structures.

In terms of ultrasound features, ultrasound images of seborrheic keratoses often include some acoustic shadows caused by the relatively strong reflectiveness of the keratotic surfaces to the ultrasound waves. Accordingly, strong reflections from the uppermost layer of the lesions and the shadows beneath this layer may be noticed on Figure 6a,b. Figure 6b presents an example of a seborrheic keratosis with parts of its surface showing different scales of reflectivity that are followed by the intensity of the acoustic shadows on the corresponding ultrasound images.

An interesting case is presented in Figure 6c. Clinically, shape and color asymmetry are visible. The dermoscopy image shows sharply demarcated borders, comedo-like openings and cerebriform structures. The ultrasound images of the depth slice of this lesion showed a suspiciously spindle-like morphology, suggesting that it may be a melanoma with an atypical surface image. However, other features, such as the presence of moderate keratosis-like shadowing and dermoscopy features were suggesting it was a seborrheic keratosis, which was then confirmed by the histopathological diagnosis.

### 3.6. Dermatofibroma

Figure 7 presents the images of a dermatofibroma, also referred to as histiocytoma. Clinically it shows as a firm pink papule with a pale central part. The dermoscopic image is characteristic showing a central whitish-pink area surrounded by a faint brownish pigmentated network. On the ultrasound images, it appears as a hypoechoic but heterogeneous lesion in the region of the dermis, with unclear borders [38].

### 3.7. Naevus

The naevus of Figure 8a was introduced to the clinical examination as an asymmetric brownish lesion. Its dermoscopic image shows a homogenous structure with no sign of malignancy. The corresponding ultrasound image presents the lesion as a hypoechoic, well demarcated dermal structure with a half-ellipse-like morphology. The information of the ultrasound image further supported the diagnosis of the dermoscopic examination.

Another lesion is presented in Figure 8b that is clinically a uniform brown, slightly elevated naevus. The dermoscopy image shows a slight asymmetry in its pigmentation. On the ultrasound image, the signs of benign naevi can be observed, such as the hypoechoic, well delimited half-ellipse-like morphology.

Figure 8c presents a compound naevus. A clinically dark brown elevated lesion, the dermoscopy image shows a papillomatous dark-brown central area with a globular pattern; the central part is surrounded by a light brown area. On the ultrasound image, the depth extent of the compound naevus is easily visible. Morphologically, the borders and shape on the ultrasound image are not as regular as those of the cases of Figure 8a,b, but the combined opto-ultrasonic information gives confidence in the diagnosis.

### 3.8. Dermatitis

Inflammatory skin diseases such as atopic dermatitis can also be followed using ultrasound imaging. It was shown in the literature that the severity of the inflammation correlates with the thickness of a subepidermal low echogenic band (SLEB) on their skin ultrasound images [10,39]. A case of atopic dermatitis is presented on Figure 9. Clinically dry skin, mild erythema with erosions and signs of lichenification are visible. The SLEB can clearly be observed on the ultrasound images of Figure 9a.

A follow-up examination is shown on Figure 9b, after two weeks of topical treatment. The differences are clearly detectable when comparing the corresponding images. On the surface, the erythematosus appearance of the skin has disappeared, and its dryness was significantly reduced. The SLEB present in the ultrasound images of the inflamed skin disappeared after treatment. The relative changes in the ultrasound reflectivity of the epidermis and dermis, as observed after the treatment, were due to the increased hydration of the skin due to the topical treatment. It is worth noting that there is also a detectable difference in the thickness of the dermis: the dermis was thicker during the inflammation than after the treatment.

### 3.9. Psoriasis

The erythematosus skin area of Figure 10a represents a psoriatic plaque. A clear sign of the inflammation is visible on the ultrasound image as a hypoechoic band beneath the epidermis, presenting the same SLEB ultrasound feature as observed in the cases of dermatitis. The relatively large thickness of the dermis, as another sign associated with the inflammation, should also be noted [13,40,41].

Figure 10b presents another case of psoriasis, on the distal end of the left ventral forearm. The lesion appears as an erythematosus, slightly scaling plaque. The lesion seen on the optical images is also observable on the ultrasound image with a clearly distinguishable SLEB, in correspondence with the observations found in the literature [13,40]. It is interesting to note that a vessel-like hypoechoic structure in the dermis is also visualized in a parallel orientation on the ultrasound image presented here.

Figure 10c presents the case of a guttate psoriasis. Small, erythematosus, slightly shiny papules of 0.5–1 cm diameter are present clinically, with signs of scaling. The dermoscopy shows whitish scales, and a small number of red dots is present. SLEB can be clearly distinguished as a marker of inflammation on the ultrasound images. However, it is not that spacious as in the images of Figure 10a,b due to the guttate nature of the inflammation.

## 4. Discussion

As described above, the *Dermus SkinScanner* gave comparable skin structure detectability to the reference device, with experience in using the devices seemingly giving some advantage in the interpretation of the images. When comparing representative images of different skin diseases, images from both ultrasound devices showed similar features. These features were also consistent with those documented in the literature.

Both devices showed melanoma with a spindle-like, fusiform shape. This is one of the typical shapes that is also reported in the literature [42,43,44,45]: one that may be used to distinguish it from other skin lesions. The irregular shape of the metastatic melanoma recorded in the current study has also been noted in the literature [43]. Since such metastases cannot be seen on the surface, they present a unique opportunity for the ultrasound to provide diagnostic assessment [46].

In the case of the BCC, several of the recordings showed hyperechoic spots within the lesion (Figure 5a,c); this feature is noted in the literature as being one of the characteristic signs of BCC [22,42,43,45,46,47], which could therefore potentially be used to differentiate the lesion from other lesion types. In addition, since the outline of BCCs is difficult to assess optically, ultrasound may serve as an aid in treatment planning [48].

The features observed in the seborrheic keratoses, dermatofiborma and naevi also corresponded with the limited number of observations reported in the literature. The strong reflectivity of the keratotic surface leads to shadows underneath [47,49]. The literature expects a dermatofibroma lesion to be ill-defined and heterogeneous, which corresponds to the ultrasound images shown on Figure 7. The naevi in Figure 8a,b were well-delimited and hyperechoic, as reported previously [46].

Finally, inflammatory skin diseases such as atopic dermatitis and psoriasis typically present a subepidermal low echogenic band (SLEB) [22] whose thickness can be related to disease severity [22,50,51]. Interestingly, the response to therapy can also be observed in both psoriasis [46,47] and dermatitis [12], as was also observed in the current study, where a shrinkage of the SLEB occurred as a response to the treatment of atopic dermatitis, as well as to changes in the echogenicity of the epidermis and dermis, and to dermis thickness.

So far, it has been demonstrated that the features observed in images recorded by the *Dermus SkinScanner* as well as by the reference device correlate well with those observed in the literature. However, the question remains of what use there may be for the multimodal, optical–ultrasound nature of the *Dermus SkinScanner* device. The combination of optical and ultrasound information has already been shown to increase the diagnostic accuracy of skin tumors [14]. Similarly, since ultrasound measures of skin inflammation have been shown to correlate well with the semi-quantitatively observed level of skin inflammation [12,22,47], even to the extent of detecting subclinical inflammation [11], the combined use of optical and ultrasound information may serve to provide better choices for personalized treatment.

In both of the above applications—improved skin cancer diagnosis and skin inflammation scoring—the ability of the *Dermus SkinScanner* to provide registration between the optical and ultrasound images is hypothesized to offer better information fusion. However, since, at the current stage of development, the optical image quality of the *Dermus SkinScanner* is visibly inferior to those of clinical and dermoscopic photos, it would seem advisable to use the optical information of the *Dermus SkinScanner* to register higher-quality optical images in addition to the ultrasound images.

The application of optical–ultrasound image fusion where the *Dermus SkinScanner* arguably has the greatest potential is in preoperative planning of skin cancer. Although the use of OCT (optical coherence tomography) for the preoperative planning of nonmelanoma skin cancer is validated by a much larger body of literature [46], ultrasound offers a more cost-effective alternative [48]. The validation of the optical–ultrasound registration methodology of the *Dermus SkinScanner* for treatment planning is the scope of future work, which validation will be helped by a future replacement of the manual scanning mechanism with motorized scanning. Another area of development is providing Doppler imaging capabilities, as this helps assess the inflammatory state of the skin and appendages, as well as vascularization (see Position Statement 3) [22]. Nevertheless, even at the current stage of development, the optical guidance of the *Dermus SkinScanner* offers the opportunity to find the maximal depth of the lesion (see Position Statement 7) [22] and images at the same location to ensure reproducibility (see Position Statement 5) [22].

## 5. Conclusions

Preliminary clinical images of a novel optical–ultrasound skin imaging device, the *Dermus SkinScanner*, were presented. The device is designed to be fully portable, cost-effective and wireless, with the aim of convenient everyday use. Although based on an essentially qualitative analysis on a relatively low sample size, recordings from a variety of common skin lesions show detectability to be comparable to an ultrasound device of similar portability, with both showing image features consistent with those reported in the literature. Since the *Dermus SkinScanner* records both optical and ultrasound images that are registered with each other, this capability presents a number of potential applications beyond the scope of the current work, including more accurate skin cancer diagnosis, skin inflammation detection and staging.

The current work presented a relatively limited number of cases and focused on qualitative presentation of the results. The collection of more cases will hopefully enable more quantitative conclusions to be drawn about the capabilities of the presented device.

To support the above, future work is warranted on two fronts. Firstly, the device needs further development for improvements to optical image quality, ultrasound position accuracy using motorization, and artefact reduction using judicious placement of the membrane. The addition of Doppler capability would also help provide additional information on inflammation and vasculature. Secondly, the potential applications of the multimodal optical–ultrasound capability should be studied even at this stage of device development, including the optical guidance of the ultrasound imaging for reproducible imaging and for finding the maximal depth of the skin tumors. Currently, one of the primary aims of development is the validation of preoperative skin cancer treatment planning using this device.

In conclusion, the first clinical results from the *Dermus SkinScanner* device presented in this current study warrant further development and investigation of the capabilities of such optical–ultrasound multimodal imaging.

## Figures and Tables

**Figure 1 diagnostics-12-00204-f001:**
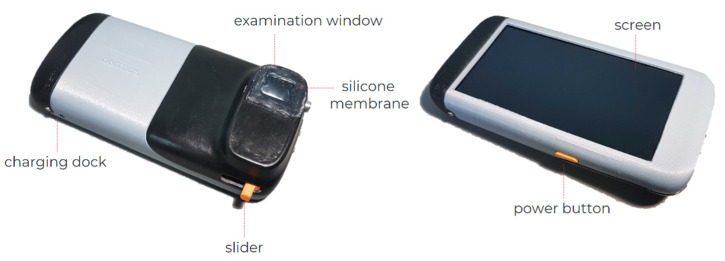
The *Dermus SkinScanner* device with its main components, including an examination window with a silicone membrane cover, screen, charging dock, power button and slider (for mechanical scanning).

**Figure 2 diagnostics-12-00204-f002:**
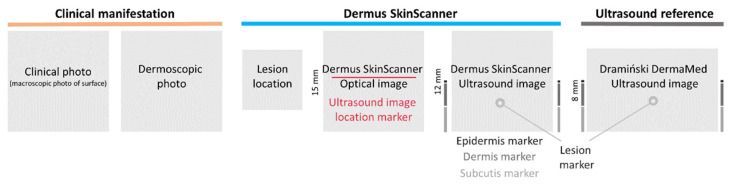
The layout in which the results of examinations are presented in this article. On the left, photo-documentation of the visual information seen by a dermatologist on a basic examination is presented: clinical (macroscopic) and dermoscopic photo of the lesion. Image information provided by the *Dermus SkinScanner* device is presented on the middle: lesion location (as marked by the investigator, on a body schematic), optical image with a red line marker representing the location of the corresponding ultrasound image slice, and the corresponding ultrasound image. The ultrasound image of the reference device is presented on the right. The primary skin layers and the lesion itself are marked on both ultrasound images as an aid for the interpretation of the images. Scaling of the images is also presented in this figure for reference (the aspect ratio is 1:1 for all of the images). The lesion markers (white stars) are placed just above the investigated lesion.

**Figure 3 diagnostics-12-00204-f003:**
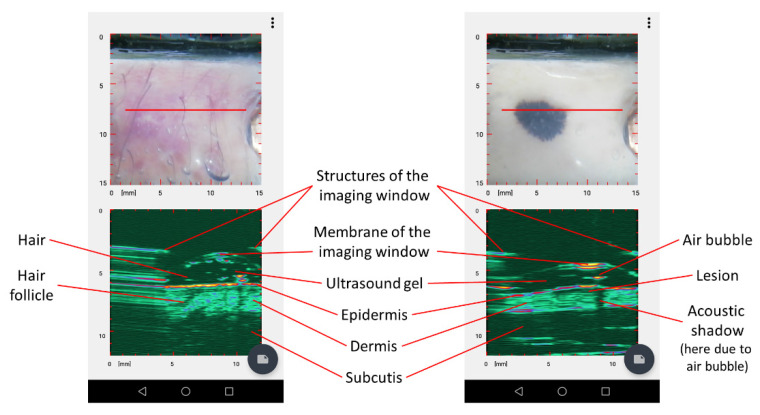
Typical elements of the ultrasound images of the *Dermus SkinScanner*. Ultrasound reflections from skin structures outline layers such as the epidermis, dermis and subcutis, and also other structures including lesions or hair follicles. These structures are preceded by reflections from the membrane and other structures of the imaging window of the device itself. The ultrasound gel filling the volume in between the imaging window membrane and the surface of the skin is reflection-free. However, reflections from hair or air bubbles may be present in this region, and they may even cause the appearance of acoustic shadows on the deeper regions of the image.

**Figure 4 diagnostics-12-00204-f004:**
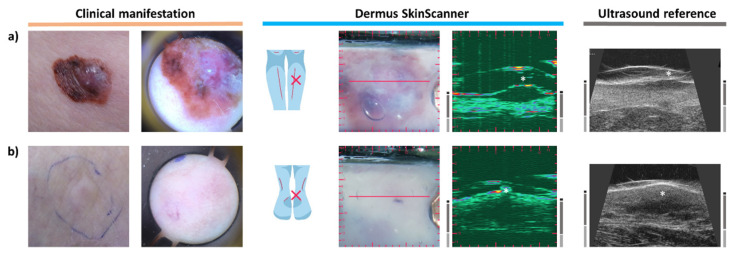
Examination images of a melanoma and of a cutaneous metastasis of melanoma (see Figure 2 for figure layout information). (**a**) The ultrasound images of the melanoma lesion show the characteristic hypoechoic spindle-like depth morphology affecting the epidermal and dermal regions. (**b**) In the case of melanoma metastasis, no characteristic features can be observed in the optical images of the skin surface. In contrast, its ultrasound images show a clearly detectable hypoechoic, heterogeneous lesion in the deeper parts of the dermis, with irregular borders and asymmetric shape.

**Figure 5 diagnostics-12-00204-f005:**
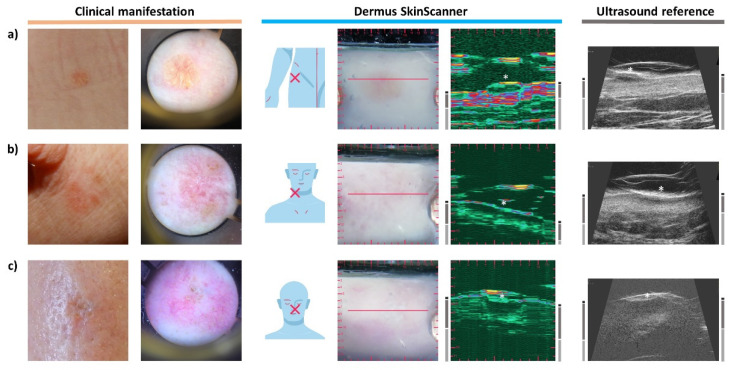
Examination images of basal cell carcinomas (BCC) (see Figure 2 for figure layout and marking information). (**a**) A slightly elevated BCC. The ultrasound images show a relatively hypoechoic (with echoes from inside), compact, but asymmetric lesion. The dermal region of this example is more hyperechoic than those shown in Figure 4; this represents the variability of skin echogenicity between patients and body locations. (**b**) A clinically flat, erythematosus BCC that was treated with cryotherapy one year before the examination. The ultrasound images show a relatively hypoechoic lesion with hypoechoic boundary regions. (**c**) A slightly sunken, scaly BCC that was treated with electrocautery therapy 3 months before the examination, being present in a location at which it is relatively hard to perform ultrasound imaging. The ultrasound images show a heterogeneously hypoechoic lesion with border irregularities.

**Figure 6 diagnostics-12-00204-f006:**
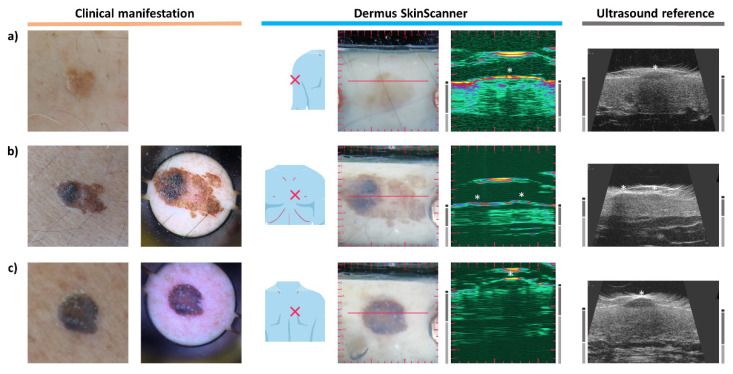
Examination images of seborrheic keratoses (SK) (see Figure 2 for figure layout and marking information). (**a**) A slightly elevated keratotic papule. The ultrasound images present the acoustic shadowing effect caused by the relatively strong reflectiveness of the keratotic surface of the lesion. (**b**) A slightly elevated SK with asymmetric shape and color. The intensity of the acoustic shadows shown on the ultrasound images follows the different scales of reflectivity of the surface of different areas of the asymmetric lesion. (**c**) An interesting case of SK. This lesion showed a suspicious spindle-like subsurface morphology (similar to those of melanoma, see Figure 4c) on the ultrasound images, with its other features, such as the presence of acoustic shadow, the relatively high echogenicity of the lesion area and the dermoscopic features (sharply demarcated borders, comedo-like openings, cerebriform structures) suggesting the correct diagnosis (SK). This example underlines the importance of employing a holistic, complete evaluation of optical and ultrasound information.

**Figure 7 diagnostics-12-00204-f007:**
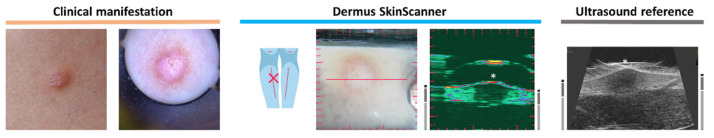
Examination images of a dermatofibroma (see Figure 2 for figure layout and marking information). The ultrasound images show a hypoechoic, heterogeneous lesion in the dermis.

**Figure 8 diagnostics-12-00204-f008:**
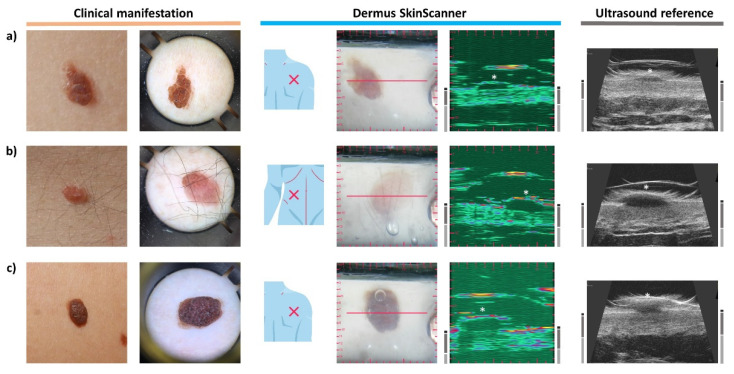
Examination images of naevi (see Figure 2 for figure layout and marking information). (**a**) An asymmetric, brownish naevus. The ultrasound images of the lesion show a hypoechoic, well delimited structure with a half-ellipse-like morphology. (**b**) The ultrasound images show a similar morphology to those of the above case. (**c**) A compound naevus. The ultrasound images show the relatively hypoechoic, well delimited lesion extending also to the non-elevated area of the dermis. Note that the acoustic shadow on the left of the *Dermus SkinScanner* ultrasound image was caused by a small air bubble inside the gel in intersection with the red line, slightly above the left side of the lesion.

**Figure 9 diagnostics-12-00204-f009:**
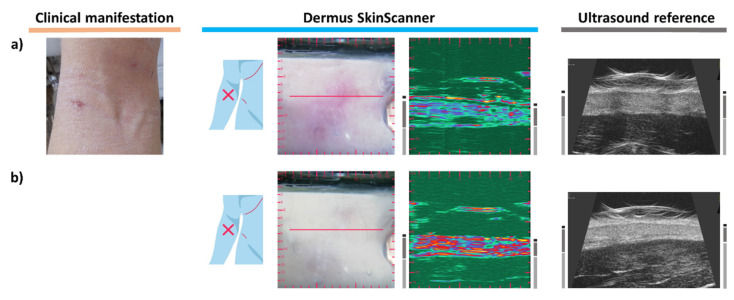
Examination images of the same skin area affected by atopic dermatitis before and after two weeks of topical treatment. (**a**) Atopic dermatitis lesion before treatment. The Subepidermal Low Echogenic Band (SLEB) can be seen on both ultrasound images. (**b**) Follow-up images after two weeks of topical treatment for atopic dermatitis. Shrinking of the SLEB as well as changes in echogenicity of the epidermis (decreased echogenicity) and dermis (increased echogenicity), due to an increased level of hydration, and slight shrinking of dermis thickness can be observed on the ultrasound images of (**b**) in comparison to those of (**a**).

**Figure 10 diagnostics-12-00204-f010:**
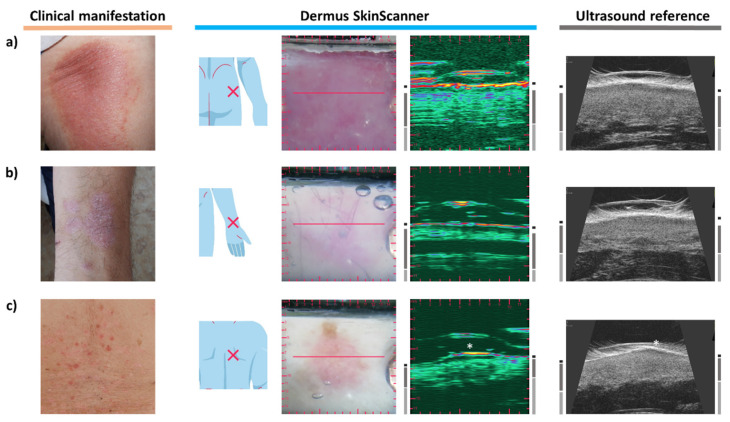
Examination images of psoriasis skin lesions (see Figure 2 for figure layout and marking information). (**a**) The ultrasound images show a clearly visible Subepidermal Low Echogenic Band (SLEB) as well as a relatively high thickness of the dermis (as signs of inflammation). (**b**) The ultrasound images show a clearly visible SLEB and a relatively thick dermis. Note the vessel-like hypoechoic structure in the dermis visualized in a parallel orientation to the image plane. (**c**) Guttate psoriasis. A SLEB with a relatively small lateral extent (compared to those of Figure 9a and Figure 10a,b) is shown on the ultrasound images.

**Table 1 diagnostics-12-00204-t001:** Detectability of the lesion and primary layers associated with the skin for the investigated and reference devices. (N = 53).

	Dermus SkinScanner			Reference Device (Dramiński DermaMed)		
	A ^1^	B ^2^	C ^3^	A ^1^	B ^2^	C ^3^
Lesion	95%	91%	91%	94%	87%	94%
Epidermis	96%	98%	92%	96%	100%	97%
Dermis	96%	91%	96%	94%	96%	96%
Subcutis	94%	85%	91%	94%	92%	96%

^1^ Investigator, on site (dermatologist). ^2^ Independent evaluator (dermatologist). ^3^ Independent evaluator (radiologist).

## Data Availability

Data are available upon reasonable request from the corresponding author.

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
