# Peer review of "Preliminary Clinical Experience with a Novel Optical–Ultrasound Imaging Device on Various Skin Lesions"

_diagnostics, 2022, doi:10.3390/diagnostics12010204_

Round 1

Reviewer 1 Report

The article seems to me very interesting, because of presentation of many skin lesions in known and new skin ultrasound imaging device. It looks interesting because small and compact. The article is well prepared, English is accurate. The figures are of good quality.

Well written and very interesting article.

Author Response

The authors are grateful to the reviewers for the time spent in carefully reading the manuscript and for the positive feedback.

Sincerely,

Gergely Csány, PhD

Reviewer 2 Report

The manuscript describes a new hand-held skin imaging ultrasound device. The subject is interesting and fits the aims & scope of this Special Session very well.

At the presentation level the manuscript is fairly well written and organised. In the introduction I missed a comparison with similar devices and related works. In particular, the following references seems relevant to the present study:

  • Chirikhina, E., Chirikhin, A., Dewsbury-Ennis, S., Bianconi, F., Xiao, P. Skin characterizations by using contact capacitive imaging and high-resolution ultrasound imaging with machine learning algorithms (2021) Applied Sciences (Switzerland), 11 (18), art. no. 8714

  • Chirikhina, E., Chirikhin, A., Xiao, P., Dewsbury-Ennis, S., Bianconi, F. In vivo assessment of water content, trans-epidermialwater loss and thickness in human facial skin (2020) Applied Sciences (Switzerland), 10 (17), art. no. 6139

  • Reginelli, A., Belfiore, M.P., Russo, A., Turriziani, F., Moscarella, E., Troiani, T., Brancaccio, G., Ronchi, A., Giunta, E., Sica, A., Iovino, F., Ciardiello, F., Franco, R., Argenziano, G., Grassi, R., Cappabianca, S. A preliminary study for quantitative assessment with HFUS (High-frequency ultrasound) of nodular skin melanoma breslow thickness in adults before surgery: Interdisciplinary team experience (2019) Current Radiopharmaceuticals, 13 (1), pp. 48-55.

  • Sciolla, B., Le Digabel, J., Josse, G., Dambry, T., Guibert, B., Delachartre, P. Joint segmentation and characterization of the dermis in 50 MHz ultrasound 2D and 3D images of the skin (2018) Computers in Biology and Medicine, 103, pp. 277-286

  • Sim, D.J.K., Kim, S.M., Kim, S.S., Doh, I. Portable skin analyzers with simultaneous measurements of transepidermal water loss, skin conductance and skin hardness (2019) Sensors (Switzerland), 19 (18), art. no. 3857

From a technical standpoint the novel contribution of the paper seems to be the description of this new skin imaging device, the main feature of which is the capability to co-register optical (frontal) and ultrasound (cross-sectional) skin images. This is indeed interesting, as such a device can find use in a range of clinical applications.

My main concern about this work, however, is that the results are essentially qualitative. This may not be as bad as it sounds (there is nothing wrong with qualitative results), but the authors should acknowledge this as a limitation of their study, and perhaps adjust title and abstract accordingly. Even the quantitative results presented in Tab. 1, should be considered as (very) preliminary ones, since the small sample size (n = 54) does not permit to draw significant conclusions.

Other remarks

Lines 97–101 The use of one specific colour palette for colorising the ultrasound image does not look as a particularly novel contribution to me. One can choose among a variety of palettes: picking one is ultimately a matter of taste.

Lines 116–123. Please provide a table to detail the number and type of lesions for the study population.

Sec. 3.1

The standard of reference (‘ground-truth’) for the results presented in Tab. 1 is unclear. Please specify.

Author Response

The authors are grateful to the reviewers for the time spent in carefully reading the manuscript and for providing constructive suggestions for improving it.

Below, you can find our responses to the comments.

To make it easily tractable, the review comments are reproduced below in italics, with our responses given in normal typeface. We have attached our revised manuscript with changes being highlighted in red. We look forward to hearing from the reviewers as to whether we have been able to address their points to their satisfaction.

Sincerely,

Gergely Csány, PhD

Responses to the comments:

The manuscript describes a new hand-held skin imaging ultrasound device. The subject is interesting and fits the aims & scope of this Special Session very well.

The authors are encouraged by this positive appraisal.

Comment 1

At the presentation level the manuscript is fairly well written and organised. In the introduction I missed a comparison with similar devices and related works. In particular, the following references seems relevant to the present study:

  • Chirikhina, E., Chirikhin, A., Dewsbury-Ennis, S., Bianconi, F., Xiao, P. Skin characterizations by using contact capacitive imaging and high-resolution ultrasound imaging with machine learning algorithms (2021) Applied Sciences (Switzerland), 11 (18), art. no. 8714
  • Chirikhina, E., Chirikhin, A., Xiao, P., Dewsbury-Ennis, S., Bianconi, F. In vivo assessment of water content, trans-epidermialwater loss and thickness in human facial skin (2020) Applied Sciences (Switzerland), 10 (17), art. no. 6139
  • Reginelli, A., Belfiore, M.P., Russo, A., Turriziani, F., Moscarella, E., Troiani, T., Brancaccio, G., Ronchi, A., Giunta, E., Sica, A., Iovino, F., Ciardiello, F., Franco, R., Argenziano, G., Grassi, R., Cappabianca, S. A preliminary study for quantitative assessment with HFUS (High-frequency ultrasound) of nodular skin melanoma breslow thickness in adults before surgery: Interdisciplinary team experience (2019) Current Radiopharmaceuticals, 13 (1), pp. 48-55.
  • Sciolla, B., Le Digabel, J., Josse, G., Dambry, T., Guibert, B., Delachartre, P. Joint segmentation and characterization of the dermis in 50 MHz ultrasound 2D and 3D images of the skin (2018) Computers in Biology and Medicine, 103, pp. 277-286
  • Sim, D.J.K., Kim, S.M., Kim, S.S., Doh, I. Portable skin analyzers with simultaneous measurements of transepidermal water loss, skin conductance and skin hardness (2019) Sensors (Switzerland), 19 (18), art. no. 3857

The authors thank the reviewer for these suggestions. In response to this comment, the last paragraph of Section 1 (Introduction) has been modified to the following (see Lines 63–77):

“A recent paper by Mlosek et al. [25] provides a valuable review of current skin ultrasound devices, noting that dedicated skin ultrasound scanners generally have higher portability than classical ultrasound scanners. Among the dedicated skin ultrasound scanners, examples of commonly used systems are the Atys Dermcup [26], Cortex DermaScan C [10], Clarius L20 HD [27], Draminski DermaMed [28], EpiScan i-200 [29,30], Vevo MD [31]. In general, the devices have a varying degree of portability, and none of them provide a viewfinder that shows in real time where the ultrasound image is taken on the skin. The aim of developing the Dermus SkinScanner was to address this need while also ensuring a high degree of portability and usability. The Dermus SkinScanner is currently a premarket device, with a novel feature of combining optical and ultrasound imaging in an integrated, handheld device. This feature aids precise positioning of the ultrasound recordings, which in turn potentially aids reproducibility of the examinations, an aim expressly stated in a recent position statement of the European Federation of Societies for Ultrasound in Medicine and Biology (EFSUMB) on dermatologic ultrasound (Position Statement 5) [22].”

Comment 2

From a technical standpoint the novel contribution of the paper seems to be the description of this new skin imaging device, the main feature of which is the capability to co-register optical (frontal) and ultrasound (cross-sectional) skin images. This is indeed interesting, as such a device can find use in a range of clinical applications.

My main concern about this work, however, is that the results are essentially qualitative. This may not be as bad as it sounds (there is nothing wrong with qualitative results), but the authors should acknowledge this as a limitation of their study, and perhaps adjust title and abstract accordingly. Even the quantitative results presented in Tab. 1, should be considered as (very) preliminary ones, since the small sample size (n = 54) does not permit to draw significant conclusions.

In response to this comment, the following modifications have been made.

The title has been changed to: “Preliminary clinical experience with a novel optical-ultrasound imaging device on various skin lesions”.

In accordance, the following sentence has been modified from

“The aim of the current work is to present the first clinical results of this device.”

to

“The aim of the current work is to present preliminary clinical results of this device.”

in the Abstract (see Line 14).

The following have been added to Section 5: Conclusions (see Lines 497–498 and Lines 505–507).

“Although based on an essentially qualitative analysis on a relatively low sample size, recordings from a variety of common skin lesions show detectability to be comparable to an ultrasound device of similar portability, with both showing image features consistent with those reported in the literature.”

“The current work presented a relatively limited number of cases and focused on qualitative presentation of the results. The collection of more cases will hopefully enable more quantitative conclusions to be drawn about the capabilities of the presented device.”

Comment 3

Lines 97–101 The use of one specific colour palette for colorising the ultrasound image does not look as a particularly novel contribution to me. One can choose among a variety of palettes: picking one is ultimately a matter of taste.

It was not the intention of the authors to claim novelty on the color scaling, but simply to present the rationale for the choice behind the color scale. To avoid misunderstanding, the sentence was changed to the following (see Lines 103–105):

“In order to provide a higher contrast and facilitate detection of skin structures, the ultrasound image employs a color scale instead of a grayscale representation.”

Comment 4

Lines 116–123. Please provide a table to detail the number and type of lesions for the study population.

The authors thank for the suggestion. Due to the relatively high number of categories compared to the case numbers, it was decided to include these case numbers in the existing paragraph (see Lines 128–132):

“The patient population included both women (38 %) and men (62 %), with mean ages of 53.7 ±21.9 and 54.1 ±19.5 years, respectively. In total, 53 lesions of 39 patients were examined. The vast majority (91 %) of the lesions were benign lesions or malignant skin tumors (including two hemangiomas, one dermatofibroma, 17 naevi, five keratoses, 12 basal cell carcinomas, two squamous cell carcinomas, eight melanomas and one cutaneous metastasis of melanoma) and a smaller proportion (9 %) consisted of inflammatory skin lesions (one dermatitis, two psoriasis, one morphea and one abscess). Regarding the locations of the lesions, the skin of several body regions has been examined, including those of the head, face, nose, upper and lower limbs, chest, abdomen and back.”

Comment 5

Sec. 3.1

The standard of reference (‘ground-truth’) for the results presented in Tab. 1 is unclear. Please specify.

The authors thank the reviewer for this suggestion. For clarification, the following sentence has been added to the end of the first paragraph of section 3.1 (see Lines 196–197).

“Please note that the only necessary standard of reference was that all imaged regions contained all four skin structures of interest.”

Reviewer 3 Report

The preprint of this manuscript has been previously uploaded at Medrxiv.org (link https://doi.org/10.1101/2021.06.28.21259325). I think this is strong research that has both theory and practical applications. The manuscript describes the research materials and methods, presents the results compared with referenced devices. In my opinion, the device description (lines 91- 101) can be given in more detail but a condensed description may be concerned with know-how or patent issues. So, if it is possible I would recommend adding this description, and while overall I think this paper can be accepted without major revision.

Author Response

The authors are grateful to the reviewers for the time spent in carefully reading the manuscript and for providing constructive suggestions for improving it.

Below, you can find our responses to the comments.

To make it easily tractable, the review comments are reproduced below in italics, with our responses given in normal typeface. We have attached our revised manuscript with changes being highlighted in red. We look forward to hearing from the reviewers as to whether we have been able to address their points to their satisfaction.

Sincerely,

Gergely Csány, PhD

Responses to the comments:

The preprint of this manuscript has been previously uploaded at Medrxiv.org (link https://doi.org/10.1101/2021.06.28.21259325). I think this is strong research that has both theory and practical applications. The manuscript describes the research materials and methods, presents the results compared with referenced devices.

The authors thank the reviewer for the positive feedback and are encouraged by this appraisal.

Comment 1

In my opinion, the device description (lines 91- 101) can be given in more detail but a condensed description may be concerned with know-how or patent issues. So, if it is possible I would recommend adding this description, and while overall I think this paper can be accepted without major revision.

In response to this suggestion, the following sentences have been added in the last paragraph of Section 2.1 (see Lines 108–110).

“The field of view of the optical image is 15 mm x 15 mm. The ultrasound image extends 12 mm laterally, with a maximum penetration depth of 10 mm. The image acquisition time is two seconds.”

Round 2

Reviewer 2 Report

The authors have addressed all my remarks